# A Biomarker Approach as Responses of Bioindicator Commercial Fish Species to Microplastic Ingestion: Assessing Tissue and Biochemical Relationships

**DOI:** 10.3390/biology11111634

**Published:** 2022-11-08

**Authors:** Xavier Capó, Merce Morató, Carme Alomar, Beatriz Rios-Fuster, Maria Valls, Montserrat Compa, Salud Deudero

**Affiliations:** Centro Oceanográfico de Baleares (IEO-CSIC), Muelle de Poniente s/n, 07015 Palma, Spain

**Keywords:** oxidative stress, microplastics, biomarkers, detoxification

## Abstract

**Simple Summary:**

Microplastic (MP) ingestion was evaluated in the gastrointestinal tract of three fish species, *Mullus surmuletus*, *Boops boops*, and *Engraulis encrasicolus*, sampled around a marine protected area (MPA). Antioxidant and detoxifying enzymes in addition to malondialdehyde levels were measured in fish liver and brain homogenates. The ingestion of MPs differed among species, with *M. surmuletus* showing the lowest microplastic ingestion of MPs and *B. boops* the highest values. An increase in MDA levels was measured in the liver of *E. encrasicolus*, as well as in the brain related to MP ingestion. An increase in CAT activity was detected in the brains of *M. surmuletus* and *B. boops.* Furthermore, GST activity in the liver of *M. surmuletus* and in the brain of *B. boops* showed an increase as a consequence of MP ingestion. MP ingestion depends on fish species and can induce an activation of detoxifying and antioxidant mechanisms that is species specific.

**Abstract:**

Plastic debris is a growing environmental problem on a global scale, as plastics and microplastics (MPs) can be ingested by marine organisms, inducing toxic effects. The aim of this study was to assess MP intake and antioxidant responses in three bioindicator species: red mullet, bogue, and anchovy (*Mullus surmuletus*, *Boops boops*, and *Engraulis encrasicolus*, respectively) for plastic contamination in the Mediterranean Sea. MP intake was assessed in the gastrointestinal tract of the fish. Further, several enzymes from both the liver and brain were analysed. The antioxidant defences, catalase (CAT) and superoxide dismutase (SOD), as well as the detoxifying enzyme glutathione-S-transferase (GST), were measured in both tissues. The acetylcholine esterase (AchE), as an indicator of neuronal damage, was measured in the brain. Malondialdehyde (MDA) was analysed as a marker of oxidative damage in the brain and liver samples. Total MP intake and MP typology differed between the three species, with *M. surmuletus* showing the lowest intake of MPs, while *B. boops* showed the highest intake of MPs. An increase in both antioxidant enzymes was evidenced in *E. encrasicolus* liver activity with respect to MP intake. In brain samples, an increase in CAT activity was found in *M. surmuletus* and *B. boops* as a consequence of MP ingestion. SOD activity in the brain increased in *B. boops* and *E. encrasicolus* that had ingested MPs. GST activity increased in the liver of *M. surmuletus*’ and in brains of *B. boops* that had ingested MPs. The intake of MPs is species related, as well as being inherently linked to the habitat they live in and being able to induce a light activation of species-specific detoxifying and antioxidant mechanisms.

## 1. Introduction

Plastic debris is a growing environmental problem on a global scale [1,2]. Once in the marine environment, plastics can fragment into smaller particles that generate microplastics (MPs), plastic particles with a diameter smaller than 5 mm [1]. MPs distribution in the sea is widespread and widely present on surface waters, across the water column, and deposited in seafloor sediments [3,4]. Due to this ubiquitous distribution, MPs can be ingested by species from several habitats and different trophic levels [5,6]. For this reason, both pelagic organisms (which live in the water column, often feeding on plankton) and benthic organisms (which live associated with the seabed and feed on organisms living along or close to the seabed) are susceptible to MP ingestion. Furthermore, due to these different feeding areas and the different distribution of MPs in sea compartments, differences in MP ingestion are expected between species [7].

MPs are manufactured from several polymers such as nylon, polypropylene, polyethylene, or polyvinyl chloride; in addition, plasticisers (such as bisphenols and phthalates) are commonly added as chemical compounds to improve the physical characteristics of plastics (durability, flexibility, etc.). These compounds, both polymers and their additives, can be toxic to marine organisms, causing a variety of negative health effects such as endocrine disruption, inflammation, and oxidative damage, and can accumulate throughout the trophic web [8].

To avoid toxic effects, all organisms present detoxifying mechanisms that convert toxic compounds into less harmful products [9]. Reactive oxygen species are commonly produced during the detoxification process of pollutants. To study this response, oxidative stress biomarkers are often used as indicators of pollution status [10]. Consequently, antioxidant enzymes, such as catalase (CAT), superoxide dismutase (SOD), glutathione reductase (GRd) and glutathione peroxidase (GPx), and biomolecule oxidation products (such as malondialdehyde (MDA) or protein carbonyl derivates) can be applied to evaluate the response of organisms to pollutants in the marine environment such as MPs, hydrocarbons, and metals [8,10]. Additionally, some studies have identified neurotoxicity as a consequence of MP ingestion; however, those have been carried out under controlled conditions [8]. Nevertheless, there is a lack of information on the effects of MPs on the antioxidant response, detoxification, and neuronal function in organisms under wild conditions.

The red mullet (*Mullus surmuletus*) is an abundant and commercially important demersal fish species with a wide distribution in the Mediterranean Sea. *M. surmuletus* preferentially inhabits sandy and muddy bottoms along the coasts and up to 400 m in depth. As a demersal species, it lives close to the bottom and feeds primarily on benthic organisms [11]. Due to its ecological and commercial importance, it is often used as a sentinel species to assess the availability of pollutants, including MPs, in sediments [6]. On the other hand, the European anchovy (*Engraulis encrasicolus*) is a small pelagic fish that is found in continental shelf waters [12]. *E. encrasicolus* inhabits the water column and feeds on small prey (usually smaller than 2 um) [13]. Several studies have revealed that anchovy ingests high amounts of MPs and therefore has been proposed as a sentinel species to monitor the abundance of MPs in the water column [14]. Finally, the bogue (*Boops boops*) is a benthopelagic species widely distributed in the Mediterranean Sea [15]. *B. boops* is a coastal fish that inhabits rocky and sandy bottoms up to 350 m depth [16]. It is a gregarious species that ascends to the surface at night and displays an omnivorous diet consuming a wide range of food items such as crustacean, porifera, seagrasses, molluscs, protozoa, and phytoplankton, both from the bottom and the water column [16,17]. Due to its wide bathymetric range of distribution and abundance, it has also been proposed as an indicator species to monitor MP pollution in the marine environment [18].

Considering this, there are two main objectives in this present study. The first is to evaluate MP ingestion in three bioindicator species, *M. surmuletus*, *B. boops*, and *E. encrasicolus*, and the second is to determine whether MP intake induces a differential physiological response in the brain and liver tissues using oxidative stress and detoxifying biomarkers found in these species.

## 2. Materials and Methods

### 2.1. Study Area

Individuals of *M. surmuletus*, *B. boops*, and *E. encrasicolus* were collected in 2020 during the annual oceanographic survey MEDITS (Mediterranean International Scientific Bottom Trawl). This survey aims at obtaining standardised basic information on the density, distribution, and demographic structure of benthic and demersal species [19]. This scientific survey was carried out during late spring around the Balearic Islands, western Mediterranean Sea, covering soft bottoms of the continental shelf (>50 m depth) and upper and middle slopes up to 800 m. Bottom trawl hauls were performed during the daytime using an experimental net (GOC37) designed for scientific purposes with a vertical opening slightly superior to most common professional gears and a cod end mesh size of 20 mm [19]. The hauls were conducted at a mean speed of 3 knots and for 20–60 min depending on the sampling depth. For this research, we selected individuals collected from hauls carried out in the surrounding waters of the marine protected area (MPA) of the Cabrera Archipelago Maritime-Terrestrial National Park (Cabrera) (Figure 1). The individuals sampled were collected from depths ranging from 80 to 140 m.

### 2.2. Biological Sampling

To study MP ingestion and associated oxidative stress response in the three study species, samples collected at each haul were immediately frozen on board at −80 °C until subsequent analysis in the laboratory. A total of 44 *M. surmuletus*, 51 *B. boops*, and 34 *E. encrasicolus* were collected. Once in the laboratory, each individual was thawed at room temperature prior to dissection. The total weight (TW, in g), eviscerated weight (EW, in g), total length (TL, in cm), and the fork length (FL) of each individual were recorded. To investigate the possible effect of MP ingestion on fish fitness or physical condition, Fulton’s condition index (CF) was calculated [20] individually following the formula: (EW in g × 100)/(FL in cm)^3^. For each sample, the gastrointestinal tract (GIT) was stored in glass Erlenmeyer flasks for digestion. To analyse oxidative stress biomarkers, the brain and liver were dissected and immediately homogenised.

### 2.3. MP Ingestion Quantification

MP ingestion was assessed in fish GIT following the harmonised biota monitoring protocol from the Plastic Busters MPA Interreg project [18]. To avoid contamination, all instruments were previously cleaned with ethanol before the start of each analysis, and researchers always wore a 100% white cotton lab coat. To isolate MPs in fish’s GIT, tissues were digested in 10% potassium hydroxide (KOH) solution (20 mL of KOH/g of tissue), as described by Dehaut [21]. Chemical digestion took place between 48 and 96 h at room temperature inside a fume hood. When all organic matter was digested, the mixture was filtered through fiberglass (FILTER-LAB fiberglass filters, pore size 1.2 μm, diameter 47 mm) with a vacuum pump and inside a fume hood to avoid airborne contamination. Once the sample was filtered, MP quantification was carried out under the stereomicroscope (Nexius Zoom 1903-S; Euromex Holland, Arnhem, The Netherlands) with optical enhancement from 6.7× to 40.5×.

Before and after filtering the samples, fiberglass filters were weighed and visually inspected under the stereomicroscope for any possible plastic contamination. Filter blanks within a glass Petri dish were used during the filtering process and visual inspection to identify potential aerial contamination. The MPs correction was applied to each sample, taking into account the number of items found in the corresponding blanks of each sample. The fish were classified into two groups according to their ingestion of MPs (no MP ingestion, and MP and ingestion).

Ingested MPs were quantified and classified according to their characteristics: size (0.3 to 1 mm, 1 to 5 mm, and 5 to 25 mm), shape (pellets, fragments, fibres, films, filaments, microbeads, foams, and others), and colour (black, blue, white/opaque, white/transparent, red, green, multicolour, and other).

### 2.4. Sample Preparation for Biomarkers

Liver samples were homogenised in 10 volumes (*w*/*v*) of 100 mM Tris-HCl buffer pH 7.5 using a small sample dispersing system (ULTRA-TURRAX^®^ Disperser, IKA (Staufen, Germany)), while the brain samples were homogenised in the same homogenisation buffer in five volumes (*w*/*v*). The homogenates were centrifuged at 9000× *g* for 10 min at 4 °C, and the supernatants were recovered and stored at −80 °C until biochemical analysis. All biochemical analyses were normalised per milligram (mg) of total protein, measured with a commercial kit (Bradford Assay: Colorimetric Protein Determination with Coomassie Blue Biorad^®^; Bio-Rad Laboratories, Hercules, CA, USA).

### 2.5. Enzymatic Activities

CAT and SOD as antioxidant defences and GST as a detoxifying enzyme were measured in liver and brain homogenates, while AChE, as a biomarker of neuronal damage, was measured in the brain of the three species. All enzyme activities were measured in homogenate supernatants with a Shimadzu^®^ UV-2401 PC spectrophotometer (Kyoto, Japan) at 25 °C. CAT activity was measured following the method described by Aebi [22]. SOD activity was determined following the method described by Flohe and Otting [23]. GST activity was determined at 340 nm using reduced glutathione (GSH) and 1-chloro-2,4-dinitrobenzene (CDNB) as substrates [24]. AChE was measured following the method described by Ellman with some modifications [25].

### 2.6. MDA Levels

MDA levels, as a marker of lipid peroxidation, were analysed using a colorimetric assay based on the reaction of MDA with a chromogenic reagent following a previously described method [10].

### 2.7. Statistical Analysis

The results of MP ingestion are expressed as means ± standard error of the mean (SEM) and for all analyses; *p* < 0.05 was considered statistically significant. A one-way analysis of variance (ANOVA) of one factor (species) was applied to assess MPs intake for each species, considering weight, size, and condition factor (CF). The Shapiro–Wilcox test was applied to assess whether the data were normally distributed among the experimental data. A one-way ANOVA of a factor (MPs ingestion) was applied to assess the effects of MP ingestion on biometric parameters and oxidative stress markers for each species separately (ANOVA, *p* < 0.05). Statistical analyses were performed using the Statistical Package for Social Sciences (SPSS v.21.0; IBM, Armonk, NY, USA) for Windows.

## 3. Results

The weight, size, and condition factor (CF) of each species are represented in Table 1. No significant differences in weight, size, or CF were found between any of the three fish species analysed with no evidence of MP ingestion in relation to those with a variable amount of MP ingestion.

The ingestion of MPs in each of the three species studied was evaluated, and the results are represented in Table 2. *M. surmuletus* showed the lowest total amount of MP ingestion (2.05 ± 0.36 MPs particle/fish), while *B. boops* showed the highest MP ingestion values (4.74 ± 1.04 MPs particle/fish).

Ingested MPs were quantified and classified according to their characteristics: size (0.3 to 1 mm, 1 to 5 mm, and 5 to 25 mm), shape (pellets, fragments, fibres, films, filaments, microbeads, foams, and others), and colour (black, blue, white/opaque, white/transparent, red, green, multicolour, and other), and the results are provided in Figure 2. In terms of the size of the MPs analysed (Figure 2A), for *M. surmuletus*, 60% of the particles analysed were within the range of 0.3 and 1 mm in length, while 35% of the particles had a size between 1 and 5 mm, and only 5% of the ingested MP particles had a size between 5 and 25 mm. In the case of *B. boops*, 87.5% of the ingested MPs had a size between 0.3 and 1 mm, and 12.5% of the ingested particles had a size between 1 and 5 mm. No particles of the size class between 5 and 25 mm were identified in the stomachs. *E. encrasicolus* showed an ingestion regarding size composition similar to *M. surmuletus* with particles between 0.3 and 1 mm (57%), 1 and 5 mm (32%), and 5 and 25 mm (10%). Ingested MPs classified according to their shape are shown in Figure 2B. In the case of *M. surmuletus*, the most common items were fibres (75%), followed by fragments (22%). No ingestion of microbeads, foams, or pellets were identified in *M. surmuletus.* In *B. boops*, fibres were the most ingested shape (88%), followed by both pellets and fragments (5.7%). No ingestion of filaments, microbeads, or foams was observed. Within the same line, in *E. encrasicolus*, the most abundant MPs were fibres (88%), followed by fragments (13%). No filament, microbead, or foam ingestion was evidenced. MPs were also classified according to their colour (Figure 2C). Regarding the shape of the MPs items, *B. boops* showed the lowest ingestion of fragments (0.36 ± 0.11 MPs particle/fish), and *E. encrasicolus* presented the highest ingestion of this type of particle (1.05 ± 0.22 MPs particle/fish). Finally, the lowest fibre ingestion was observed in *M. surmuletus* (1.60 ± 0.28 MPs particle/fish), while the highest ingestion of fibres was observed in *B. boops* (4.38 ± 1.01 MPs particle/fish). Multicolour MPs were the most abundant in *M. surmuletus* (38%), while no red MPs were found. In *B. boops*, most of the MP particles ingested were blue (47.5%), followed by multicolour MPs (11.8%) and white/transparent (14.5%). Finally, in *E. encrasicolus*, blue was also the most abundant colour comprising the MPs ingested (38%), followed by transparent particles (35.7%).

In terms of biomarker responses from the ingestion of MPs in each species, CAT and SOD values in brain and liver samples from the three studied species were measured as markers of antioxidant status, and no effects of MP ingestion on CAT and SOD activity in the liver were observed in either *M. surmuletus* or *B. boops* (Table 3). On the other hand, an increase in CAT and SOD in liver activity was observed in *E. encrasicolus* that had ingested MPs. In brain samples, an increase in CAT activity was found as a consequence of MP ingestion in *M. surmuletus* and *B. boops*, but no effects of MP ingestion were observed in brain samples of *E. encrasicolus.* SOD activity in the brain increased as a consequence of MP ingestion in both *B. boops* and *E. encrasicolus*. However, no variation in SOD activity was observed in the brain of *M. surmuletus* as a consequence of MP ingestion.

GST activity in both tissues was analysed for the three studied species as a marker of the detoxification process (Figure 3). Among the three species studied that had ingested MPs, only *M. surmuletus* individuals showed an increase in liver GST activity (ANOVA, *p* > 0.05). Regarding the brain samples, an increase in GST activity was only detected in *B. boops* (ANOVA, *p* > 0.05).

AChE activity analysed in the brain to determine neuronal damage showed no differences related to MP ingestion in any of the three species (Figure 4). Similarly, mean MDA values for each of the three species with intake and absence of MP showed no significant differences (Table 4). No effects of MP ingestion on MDA levels in the brain or liver were observed, neither in *M. surmuletus* nor in *B. boops.* However, a significant increase (ANOVA, *p* > 0.01) in MDA levels in liver was detected in *E. encrasicolus* that was related to the ingestion of MPs.

## 4. Discussion

MPs are ubiquitous in the marine environment and abundant in the water column, along the surface, in shallow coastal sediments, and even at deeper abyssal depths [4]. The distribution and accumulation of MPs in the marine environment is modulated by several factors such as wind, hydrography, anthropogenic activities, and size of MPs [18]. Considering their wide distribution in marine ecosystems, the availability of MPs to several marine organisms such as pelagic and benthic fish, molluscs, jellyfish, cetaceans, marine turtles, and seabirds is a reality, and ingestion of these particles has been reported in multiple species belonging to these groups [6,26].

Previous studies highlighted the Mediterranean as a hotspot for marine diversity and plastic pollution [27], while other authors identified specific areas where marine species are at high risk of ingesting plastic, primarily in coastal areas [28]. In the Balearic Islands (western Mediterranean), the overlap between 54 species and seafloor marine litter suggests that the species inhabiting the region along the south western coast of Mallorca were shown to be more exposed to plastic pollution [27].

### 4.1. MP Intake

In this study, we have reported the ingestion of MPs for three different species with different habitats and feeding traits. Different ingestion rates for each species, in addition to differences in MP shape and colour particle ingestion between the three species analysed, have been observed in the study area. In the case of *M. surmuletus*, 70.5% of the individuals analysed had ingested MPs, 95% of *E. encrasicolus*, and 64% of *B. boops*. In the case of the number of MPs particles ingested, *B. boops* presented the highest total MP intake, followed by *E. encrasicolus*. Considering the type of microplastics ingested, *E. encrasicolus* showed the highest MP fragment ingestion, while *B. boops* was the species with the highest fibres ingestion. On the other hand, *M. surmuletus*, which primarily feeds on small benthic crustaceans, was the species with the lowest total amount of MP ingestion. These ingestion patterns suggest a higher presence of MPs in the water column.

MP ingestion in the three studied species has already been quantified in the study region. According to previous data from *M. surmuletus*, a plastic occurrence of 27% and a mean value of 0.42 ± 0.04 MPs/fish [6] was observed in individuals from the same study area, and these values were much lower than those obtained in the present study. Given that seafloor habitats are considered sinking areas of MPs, some authors suggest that species feeding on the seafloor have higher ingestion values than pelagic species [5]. However, this is not the case when considering the three species analysed in this study, as lower ingestion values were reported for the bottom-dwelling fish *M. surmuletus*, compared to the pelagic and benthopelagic feeder fish *E. encrasicolus* and *B. boops*, respectively. Ingestion of MPs in *E. encrasicolus* in this study was higher than previously reported values in the same area, since the ingestion of anthropogenic particles in fish was observed with a very low frequency along the Spanish peninsula coast with percentage occurrence values ranging from 6.67% to 14.28% and mean values from 0.07 ± 0.26 particles/individual to 0.18 ± 0.02 particles/individual [14,28]. Regarding *B. boops*, values obtained in this study were in the range of those previously reported in the Balearic Sea, with a mean value of 3.75 ± 0.25 MPs/individual. Comparisons between studies conducted in the same area were highlighted as having high variability between and within species [27].

Variability in MP ingestion within the same species could be attributed to differences in diet composition related to prey availability [29] and also to biological factors such as fish size, as plastic ingestion can increase with fish length [28]. The three studied species were selected owing their different feeding strategies. *M. surmuletus* is a bottom dwelling species foraging on benthic invertebrates, while *E. encrasicolus* is a pelagic species that feeds on planktonic organisms. *B. boops* is a benthopelagic species that feeds both near the bottom and in the water column. Taking into account the different feeding strategies, the results of this study highlight that different fish species can indirectly ingest items that are similar in size and shape to prey organisms. In relation to the shape of the MPs ingested, different results were observed compared to previous studies conducted in the Balearic Islands, as [14] reported only fibre ingestion for *B. boops*, while in this study, we also evidenced an ingestion of pellets and fragments but in low abundance, which could have been due to temporal differences in the presence of microplastics in the environment. Previous studies had also analysed the shape of ingested MPs in *M. surmuletus* [6], and in the latter study, it was reported that 97% of the ingested MPs by *M. surmuletus* were fibres, while our results showed lower fibre ingestion (75%) and the presence of MP fragments (22%). In terms of colours, blue and white MPs were the predominant colours in *B. boops* and *E. encrasicolus*, which are the two species with pelagic feeding, while in *M. surmuletus*, a greater diversity in MPs colours has been evidenced; these results are similar to those previously reported in these species [6,30,31].

This study reveals that MPAs are not free from plastic pollution, and in some cases, this pollution is higher than in high anthroponised areas [32]. Consequently, ecological and biological traits, as well as the proximity of sampling areas to potential sources of plastics, including MPA, must be considered when comparing MP ingestion. In addition, an intraspecific and multiple-specific approach, covering different depth strata, habitats, and feeding strategies, should be assessed [27].

### 4.2. Biomarker Response

At the species level, a differential physiological response of the liver and brain was observed as being associated with MP ingestion. The liver plays an important role in the detoxification processes of xenobiotics, and consequently, this tissue is often used as an indicator of the degree of damage induced by pollutants, particularly in the case of MPs [8]. Similarly, the brain is a tissue that is used to determine the neurotoxic effects of pollutants. In this sense, neurotoxic effects and increased oxidative stress biomarkers have already been associated with MP ingestion [8,33]. However, our results show slight changes in biochemical markers as a consequence of MP ingestion. At this point, it is interesting to highlight that the vast majority of previous studies in which biochemical markers showed higher differences between individuals that have ingested MPs and individuals without MP ingestion were carried out under laboratory conditions with continuous plastic exposure [10].

### 4.3. Antioxidant Response

Antioxidant and detoxifying enzyme activities (CAT, SOD, GRd, GPx, and GST) and oxidative damage markers (MDA, protein carbonyl derived) are traditionally used as indicators of the degree of stress associated with pollution [34]. In this study, we evaluated the antioxidant response (CAT and SOD), detoxifying activity (GST), oxidative damage markers (MDA), and neuronal damage (AchE in brain samples) in the three species. In this study, slight changes in the activities of biochemical markers were detected as a consequence of MP ingestion. Different responses to MP ingestion were observed, depending on the species and tissue analysed. A low increase in *M. surmuletus* liver GST activity was detected, whereas an increase in CAT activity was observed for *E. encrasicolus* that had ingested MPs. In contrast, no effects on any antioxidant or detoxifying enzymes were observed in the liver of *B. boops* as a consequence of MP ingestion. In addition, changes in oxidative damage markers were also very slight in the present research. Only *E. encrasicolus* revealed an increase in liver MDA levels as a consequence of MP ingestion. These results are consistent with previous studies in which practically no effects were found with respect to oxidative stress markers as a consequence of MP ingestion in wild fish species [6]. Furthermore, in our study, antioxidant and detoxifying markers (CAT, SOD, and GST), neuronal damage markers (AChE), and oxidative damage markers (MDA) in the brain did not provide clear evidence of the effect of MP ingestion on brain functionality.

Previous studies under controlled conditions have evidenced important harmful effects of MP ingestion in the brain of fish species [8]. Similarly, another study under controlled conditions evidenced neurotoxicity effects as a consequence of MP ingestion in sea bass (*Dicentrarchus labrax*) [8]. On the other hand, we found changes in oxidative stress markers in the brain as consequence of MP ingestion. In the case of *M. surmuletus*, we only identified an increase in CAT brain activity associated with MP ingestion, while the other markers of oxidative damage did not show any effects of MP ingestion. According to *B. boops*, the brain did show an increase in GST, CAT, and SOD activities in individuals with MP ingestion in comparison to fish without MP ingestion. However, no effects of MP ingestion on markers of brain oxidative damage were detected in the study area.

## 5. Conclusions

Finally, biomarkers measured in the brain of *E. encrasicolus* were modulated by MP ingestion, as an increase in SOD activity was associated with MP ingestion, and simultaneously an increase in oxidative damage markers was also identified in those individuals that had ingested MPs. In conclusion, characteristics of MPs ingested depends on fish species as well as habitat of each species. MP ingestion can induce a slight activation of detoxifying and antioxidant mechanisms, and this activation is species specific. The results of this study indicate that there was no correlation between MP ingestion and antioxidant or detoxifying enzyme activities in the liver and brain of fish species measured in wild conditions. However, some enzyme activities such as CAT and SOD, as well as SOD and AChE, were correlated, depending on the species and tissues. The overall results from this study highlight how biomarkers are activated in liver and brain tissues in relation to MP ingestion in wild bioindicator species. Considering these results, there were a few limitations from this study. For example, samples were collected during specific dates (during summer months), and locations providing baseline data and biannual surveys for these areas for the three bioindicator species would provide a long time series for the region. Additionally, future studies should consider not only seasonality but also the ontogeny of each of the studied species considered. Here, we provided the results for adults, and the oxidative response may be different in juveniles or even potentially alter their development. Overall, more research is needed to further monitor the toxicity and biological implications of the bioavailability of MPs in marine ecosystems on the overall health of commercially important bioindicator species.

## Figures and Tables

**Figure 1 biology-11-01634-f001:**
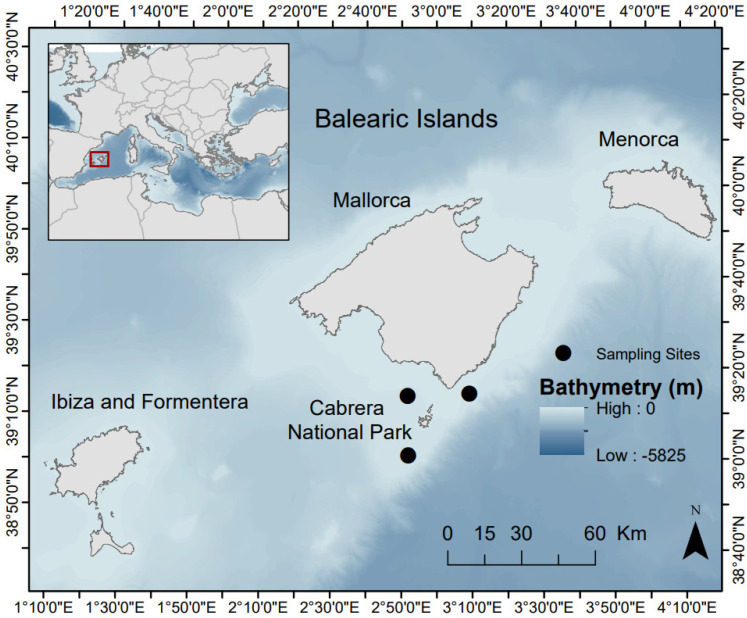
Map of the locations of the sampling sites in the surrounding waters of the Cabrera Archipelago Maritime-Terrestrial National Park. Inset map indicates the location of the sampling area in the Balearic Islands in the western Mediterranean Sea.

**Figure 2 biology-11-01634-f002:**
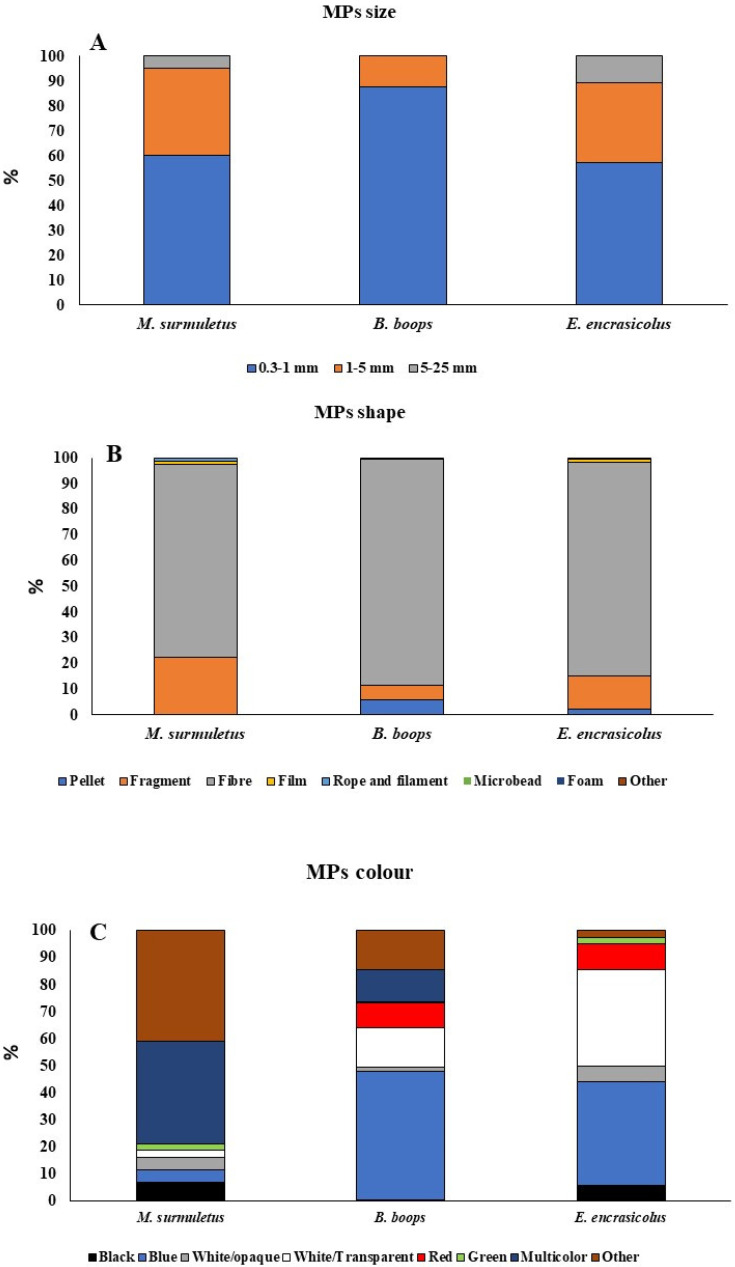
Relative abundance of different types of MPs ingested in commercial fish. (**A**) MPs size (**B**) MPs shape (**C**) MPs colour.

**Figure 3 biology-11-01634-f003:**
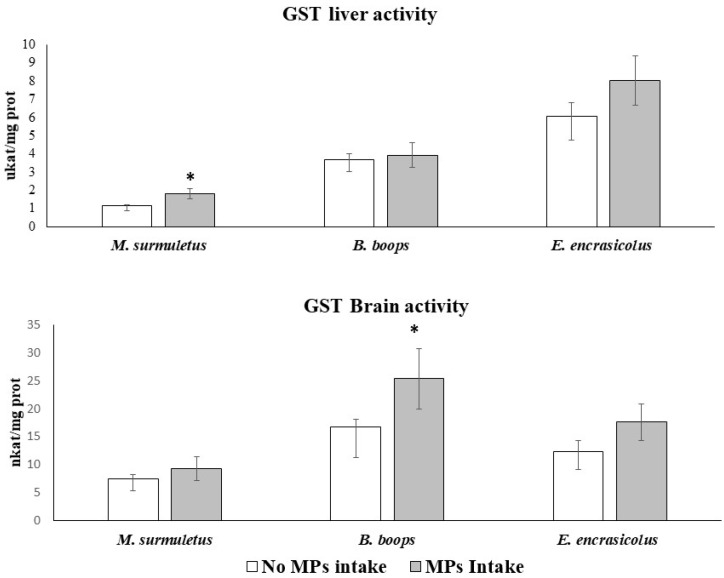
GST activity in both tissues in the three study species. * indicates significant differences with respect to no MP ingestion. Brain and liver of 44 *M. surmuletus*, 51 *B. boops*, and 34 *E. encrasicolus* were used to perform this determination. Statistical analysis: one-way ANOVA. *p* < 0.05.

**Figure 4 biology-11-01634-f004:**
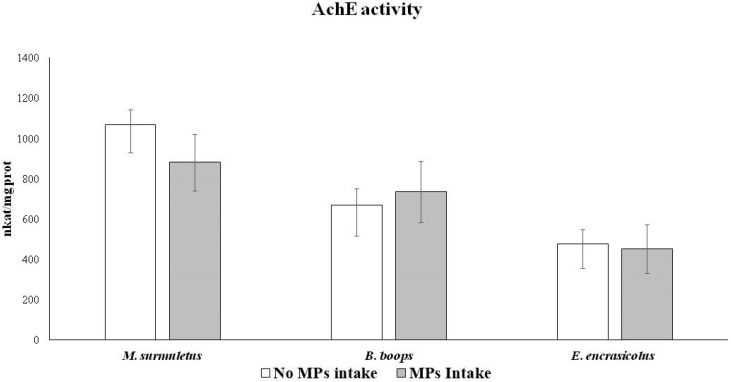
AChE activity in brain in the three study species. Brain and liver of 44 *M. surmuletus*, 51 *B. boops*, and 34 *E. encrasicolus* were used to perform this determination. Statistical analysis: one-way ANOVA. *p* < 0.05.

**Table 1 biology-11-01634-t001:** Biometric characteristics of each studied species.

Species	Group	Length (cm)	Weight (g)	Condition Factor (CF)
*Mullus surmuletus*	No MP intake	18.1 ± 0.37	65.8 ± 3.40	1.45 ± 0.07
	MP intake	18.1 ± 0.45	70.4 ± 4.26	1.44 ± 0.05
*Boops boops*	No MP intake	18.1 ± 0.39	57.7 ± 3.46	1.11 ± 0.02
	MP intake	17.6 ± 0.71	53.6 ± 5.26	1.04 ± 0.04
*Engraulis encrasicolus*	No MP intake	8.04 ± 0.39	10.4 ± 0.19	0.64 ± 0.02
	MP intake	8.88 ± 0.40	10.8 ± 0.13	0.64 ± 0.03

*p* < 0.05. Statistical analysis: one-way ANOVA.

**Table 2 biology-11-01634-t002:** Microplastic intake for each of the studied bioindicator species.

Species	Fragments	Fibres	Total MPs
*Mullus surmuletus*	0.48 ± 0.18	1.60 ± 0.28	2.05 ± 0.36
*Boops boops*	0.36 ± 0.11	4.38 ± 1.01 *	4.74 ± 1.04 *
*Engraulis encrasicolus*	1.05 ± 0.22 *^,#^	2.68 ± 0.33	3.74 ± 0.41

* Difference with respect to *M. surmuletus*, ^#^ difference with respect to *B. boops.* Statistical analysis: one-way ANOVA. *p* < 0.05.

**Table 3 biology-11-01634-t003:** Antioxidant enzymes damage in the liver and in the brain in three fish species.

	Plastic Intake	*Mullus surmuletus*	*Boops boops*	*Engraulis encrasicolus*
**Liver**				
**CAT** **(mK/mg prot)**	No intake	939 ± 65.7	954 ± 77.5	966 ± 103
	Intake	882 ± 108	848 ± 146	1582 ± 179 *
**SOD** **(pkat/mg prot)**	No intake	52.8 ± 2.95	50.3 ± 3.15	25.8 ± 2.21
	Intake	53.1 ± 3.01	45.6 ± 5.38	33.4 ± 2.53 *
**Brain**				
**CAT** **(mK/mg prot)**	No intake	154 ± 16.2	123 ± 16.2	697 ± 155
	Intake	247 ± 61.7 *	219 ± 27.3 *	465 ± 177
**SOD** **(pkat/mg prot)**	No intake	14.9 ± 0.88	16.4 ± 1.58	36.2 ± 3.26
	Intake	13.8 ± 1.4	30.4 ± 3.79 *	50.2 ± 2.59 *

* indicates significant differences with respect to no MPs intake. *p* < 0.05. Statistical analysis: one-way ANOVA.

**Table 4 biology-11-01634-t004:** Oxidative damage in the liver and in the brain in three fish species.

	Plastic Intake	*Mullus surmuletus*	*Boops boops*	*Engraulis encrasicolus*
**Liver**				
**LPO** **(nmols/mg of prot)**	No intake	0.66 ± 0.06	0.92 ± 0.06	0.89 ± 0.16
	Intake	0.49 ± 0.09	0.74 ± 0.11	0.88 ± 0.07
**MDA** **(nmols/mg of prot)**	No intake	1287 ± 136	1876 ± 203	1343 ± 135
	Intake	1204 ± 295	1622 ± 322	2872 ± 318 *
**Brain**				
**LPO** **(nmols/mg of prot)**	No intake	0.80 ± 0.07	1.12 ± 0.06	2.24 ± 0.11
	Intake	0.99 ± 0.23	1.12 ± 0.09	3.56 ± 0.43 *
**MDA** **(nmols/mg of prot)**	No intake	887 ± 61.4	875 ± 51.9	550 ± 72.3
	Intake	1021 ± 242	1307 ± 118	1042 ± 203 *

* indicates significant differences with respect to no MPs intake. *p* < 0.05. Statistical analysis: one-way ANOVA.

## Data Availability

Researchers wishing to access the data used in this study can make a request to the corresponding author.

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
