# Peer review of "A Biomarker Approach as Responses of Bioindicator Commercial Fish Species to Microplastic Ingestion: Assessing Tissue and Biochemical Relationships"

_biology, 2022, doi:10.3390/biology11111634_

Round 1

Reviewer 1 Report

I revised the paper intitled: A biomarker approach to responses of bioindicator commercial fish species to microplastic ingestion: assessing tissue and biochemical relationships. This is an extensive and quite interesting study, with plenty of valuable results.

Full formatting is needs, as the authors must follow exactly the formatting indicated in the authors' instructions, since nothing is in accordance with the intended format.

An English revision is needed in some points, although the writing is easy to read and done in a clear and objective way. Furthermore, it seems to me that the purpose of this study is clear and well defined. T I found a limited number of articles dealing with this topic on the ISI web and other online search engines. Therefore, I therefore admit novelty and contribution.

Conclusions must be reinforced, clearly indicating what contribution these results can make in this area of research and which the study brought as a novelty.

All graph legends must indicate the value of n.

 Additionally other revisions suggested:

Line 43: I think that “dispose of” is not the correct term

Line 48: (GPx) instead of GPx)

Line 68: species

Line 127 and 171: -80°C instead of -80ºC

Line 172: please indicate the kit name

Author Response

Review Report 1

I revised the paper intitled: A biomarker approach to responses of bioindicator commercial fish species to microplastic ingestion: assessing tissue and biochemical relationships. This is an extensive and quite interesting study, with plenty of valuable results. Full formatting is needs, as the authors must follow exactly the formatting indicated in the authors' instructions, since nothing is in accordance with the intended format.

We are grateful for the revision provided by Reviewer #1. Regarding the format of the manuscript, we have followed the journal’s format as they do not have any strict formatting requirements as long as we comply with the mandatory sections: Abstract, Keywords, Introduction, Materials & Methods, Results, Conclusions, Figures and Tables with Captions, Funding Information, Author Contributions, Conflict of Interest and other Ethics Statements. We have addressed the sections for: "Institutional Review Board Statement", "Informed Consent Statement", "Data Availability Statement".

An English revision is needed in some points, although the writing is easy to read and done in a clear and objective way.

Following the reviewer’s comments, the new version has been carefully revised by a native English speaker.

Furthermore, it seems to me that the purpose of this study is clear and well defined. T I found a limited number of articles dealing with this topic on the ISI web and other online search engines. Therefore, I therefore admit novelty and contribution.

Conclusions must be reinforced, clearly indicating what contribution these results can make in this area of research and which the study brought as a novelty.

Following the reviewer's recommendations, we have rewritten the conclusions.

All graph legends must indicate the value of n.

We have included the n value but only in the graph legend of biomarkers. In the case of MPs characteristics, we have not added the n because taking into account, the three species, and the great variety of characteristics of the plastics would make the graph legend unintelligible.

 Additionally other revisions suggested:

Line 43: I think that “dispose of” is not the correct term

We have corrected according to reviewer’s suggestion.

Line 48: (GPx) instead of GPx).

We have corrected according to reviewer’s suggestion.

 Line 68: species

We have corrected according to reviewer’s suggestion.

Line 127 and 171: -80°C instead of -80ºC

We have corrected according to reviewer’s suggestion.

Line 172: please indicate the kit name

We have corrected according to reviewer’s suggestion.

Reviewer 2 Report

The manuscript focus on a biomarker approach to responses of bioindicator commercial fish species to microplastic ingestion. I think it's novel and excellent work, but many questions need to be revised and corrected.

General comments:

1.There is no doubt that the authors have done a lot of work, but the presentation is too simple.

2.Abstract section: relevant analysis results (simple introduction and significance, methods, results, and conclusion) need to be presented, and the content is too little overall.

Special comments:

3.Line 73-76: This part can be refined in the first sentence of the abstract section.

4.Line 95: “Figure 1. Map of the locations of the sampling ......” It should be at the bottom of the Figure, which is the basic requirement of the journal (Biology). In addition, it is also necessary to place a wider range of graph in the Figure 1. Otherwise, the audiences do not know where it is?

5.Line 258: the bottom of the Figure.

6.Discussion section: Supplement subtitles to make the content clearer.

7.Conclusion section: Add the limitations of the study.

Author Response

Review Report 2

The manuscript focus on a biomarker approach to responses of bioindicator commercial fish species to microplastic ingestion. I think it's novel and excellent work, but many questions need to be revised and corrected.

General comments:

1.There is no doubt that the authors have done a lot of work, but the presentation is too simple.

We have revised the presentation of the manuscript in order to improve it.

2.Abstract section: relevant analysis results (simple introduction and significance, methods, results, and conclusion) need to be presented, and the content is too little overall.

We have rewritten the abstract, an added more information.

Special comments:

3.Line 73-76: This part can be refined in the first sentence of the abstract section.

We have modified according to reviewer’s recommendation

4.Line 95: “Figure 1. Map of the locations of the sampling ......” It should be at the bottom of the Figure, which is the basic requirement of the journal (Biology). In addition, it is also necessary to place a wider range of graph in the Figure 1. Otherwise, the audiences do not know where it is?

We have modified the caption for Figure 1 and we have included the whole western Mediterranean Sea for reference in the inset map.

5.Line 258: the bottom of the Figure.

We have corrected according to reviewer’s suggestion

6.Discussion section: Supplement subtitles to make the content clearer.

We have added subtitles to discussion section

7.Conclusion section: Add the limitations of the study.

 We have added the following sentence as study limitation.

“Given samples were collected during specific dates and locations, annually surveys for these areas for the three bioindicator species would provide to provide a long time series for the region, as for now the results from this study provide baseline data for the region”.

In addition, please also revise your manuscript according to the journal request:

  1. Please add the "Simple Summary" section in front of the "Abstract" section. Here I attached the template of Biology for your reference.

We have added the “simple Summary” according journal template.

  1. 2. Please complete the back matter ("Institutional Review Board Statement", "Informed Consent Statement", "Data Availability Statement") according to the

new template and the following link:

We have completed according to journal requirements.

  1. The self-citation rate of your paper is too high (43%). Our requirement is lower than 15%. Please add new references or remove references authored by you to reduce the self-citation rate to below 15%.
  • We have removed and changed some citations, however we have to keep a few more citations as the studies in the study area are carried out by contributing authors to this publication.

Round 2

Reviewer 2 Report

1.Line 90-94: The objectives and significance of this research are not very clear and need to be revised.

2.Line 112: Small maps need to add a larger range. At present, this range is not clear.

3.Materials and Methods section: Some descriptions are unnecessary. The author needs to carefully modify this part. Perhaps this part is the template of the author's laboratory, but this is necessary to revise it.

Line 119: “Once in the laboratory, each individual was thawed at room......

Line 131: Researchers always wore a 100% white cotton lab coat......

4.Line 254: (ANOVA, p=0.005) is error? It should be (ANOVA, p<0.005). *** is p<0.01, ** is p<0.05, and * is p<0.1.Statistical common sense like this needs to be revised, e.g., Table 2, Line 242, 244, 248.

5.Line 379: Study limitation:...... This can be placed in the last paragraph of the conclusion. The conclusion includes the article summary, research limitations and future work.

Author Response

Review  report 2.

1.Line 90-94: The objectives and significance of this research are not very clear and need to be revised.

We have revised the objectives from the introduction section and have modified to the following for clarity:

‘Considering this, there are two main objectives in this present study. The first is to evaluate MP ingestion in three bioindicator species, M. surmuletus, B. boops and E. encrasicolus and the second is to determine whether if MPs intake induces a differential physiological response in the brain and liver tissues using oxidative stress and detoxifying biomarkers found in these species.’

2.Line 112: Small maps need to add a larger range. At present, this range is not clear.

We have changed the map according to reviewers’ suggestion and the entirety of the Mediterranean Sea is now visible in the inset map.

3.Materials and Methods section: Some descriptions are unnecessary. The author needs to carefully modify this part. Perhaps this part is the template of the author's laboratory, but this is necessary to revise it.

We have the revised material and methods section. However, authors consider that due to the complexity of some analyses, all information is necessary to ensure a correct replication of the analyses performed.

Line 119: “Once in the laboratory, each individual was thawed at room......”.

Authors consider that this information is necessary. In this case, the fact that samples were thawed at room temperature, allow the author to measure biometric data in a much more accurate way. Additionally, this facilitated a better dissection of the tissues that were analysed

Line 131: “Researchers always wore a 100% white cotton lab coat…...”

In this case authors also consider that this information is necessary, the use 100% white cotton lab coat allowed reduce potential sources of contamination

4.Line 254: “(ANOVA, p=0.005)” is error? It should be (ANOVA, p<0.005). *** is p<0.01, ** is p<0.05, and * is p<0.1. Statistical common sense like this needs to be revised, e.g., Table 2, Line 242, 244, 248.

We have not found this information in line 254. In line 394 it is indicated that p value for ANOVA analysis was 0.005, and this information is right.  

5.Line 379: “Study limitation:......” This can be placed in the last paragraph of the conclusion. The conclusion includes the article summary, research limitations and future work.

We have placed study limitation in the last paragraph of the conclusion according reviewer’s suggestion.

Round 3

Reviewer 2 Report

1.The authors need to confirm whether the statistical symbols in the manuscript are correct, p=0.005, right?

E.g., line 181, p < 0.05, represents significance in statistically, right. But, line 243, p=0.02, line 244, p=0.04, line 254, p=0.005. These p-value is equal to certain value.

2.About the map. If the reader of the journal is not from a Mediterranean country, he will not be aware of this study area, therefore, it is hoped that the author will put a larger map of the area. Ref.  (https://doi.org/10.3390/biology11101468) or (https://doi.org/10.3390/biology11050732).

Author Response

Reviewer’s response

1.The authors need to confirm whether the statistical symbols in the manuscript are correct, p=0.005, right?

The p-value considered statistically significant is ≤0.05 as indicated in all figure captions.

E.g., line 181, p < 0.05, represents significance in statistically, right. But, line 243, p=0.02, line 244, p=0.04, line 254, p=0.005. These p-value is equal to certain value.

In the specific cases commented on by the reviewer, the p-value for each ANOVA analysis was the value indicated  in brakets, which in all cases is less than 0.05 which is the value p-value considered statistically significant.

2.About the map. If the reader of the journal is not from a Mediterranean country, he will not be aware of this study area, therefore, it is hoped that the author will put a larger map of the area. Ref.  (https://doi.org/10.3390/biology11101468) or (https://doi.org/10.3390/biology11050732).

We have changed the map according to reviewers’ suggestion and the entirety of the Mediterranean Sea is now visible in the inset map.
